# Transcription Factors in *Aureobasidium* spp.: Classification, Regulation and a Newly Built Database

**DOI:** 10.3390/jof8101096

**Published:** 2022-10-17

**Authors:** Guang Yang, Yuhan Wang, Yaowei Fang, Hongjuan Mo, Zhihong Hu, Xiaoyue Hou, Shu Liu, Zhongwei Chen, Shulei Jia

**Affiliations:** 1Jiangsu Key Laboratory of Marine Bioresources and Environment, Jiangsu Ocean University, Lianyungang 222005, China; 2Jiangsu Marine Resources Development Research Institute, Jiangsu Ocean University, Lianyungang 222005, China; 3Co-Innovation Center of Jiangsu Marine Bio-Industry Technology, Jiangsu Ocean University, Lianyungang 222005, China; 4College of Food Science and Engineering, Jiangsu Ocean University, Lianyungang 222005, China; 5National Marine Environmental Monitoring Center, Dalian 116023, China; 6Institute of Microbiology, Chinese Academy of Sciences, Beijing 100101, China

**Keywords:** *Aureobasidium* spp., transcription factors, PFAM families, classification and regulation, retrieval database

## Abstract

Transcription factors (TFs) can regulate the synthesis of secondary metabolites through different metabolic pathways in *Aureobasidium* spp. In this study, a set of 16 superfamilies, 45 PFAM families of TFs with the DNA-binding domains, seven zinc finger families and eight categories of the C2H2 TFs have been identified in *Aureobasidium* spp. Among all the identified TFs, four superfamilies and six PFAM families are the fungal-specific types in this lineage. The Zn2Cys6 and fungal-specific domain regulators are found to be overwhelmingly predominated, while the C2H2 zinc finger class comprises a smaller regulator class. Since there are currently no databases that allow for easy exploration of the TFs in *Aureobasidium* spp., based on over 50 references and 2405 homologous TFs, the first TFs pipeline—the *Aureobasidium* Transcription Factor Database (ATFDB)—has been developed to accelerate the identification of metabolic regulation in various *Aureobasidium* species. It would be useful to investigate the mechanisms behind the wide adaptability and metabolite diversity of *Aureobasidium* spp.

## 1. Introduction

*Aureobasidium* is a genus of ascomycete in the family Saccotheciaceae that is found worldwide in water, soil, wood, rock, saline habitats, limestone, coastal waters, deep sea, marine sediments from the Antarctic, desert, honeycomb and mangrove ecosystems [1]. This genus is characterized by a high capacity for adaptation to a complex and changing living environment. Diverse strains of *Aureobasidium* spp. produce a variety of secondary metabolites, including polymalate (PMA), pullulan, liamocins, siderophores, melanin, polyunsaturated fatty acids (PUFA), massoia lactone, intracellular lipids, gluconic acids (GA), fructooligosaccharides (FOSs) and various enzymes, indicating that they have potential applications in biotechnology [2,3,4,5,6]. The biosynthesis of secondary metabolites in *Aureobasidium* spp. is intimately related to the distinct signal pathways, the whereabouts of carbon metabolic flow and regulation of various transcription factors (TFs) [7,8]. Transcription factors are recognized to have a vital role in influencing the levels of pathway gene expression, hence regulating flow via secondary metabolic pathways [9,10].

Until now, numerous different types of transcription factors have been intensively studied in yeast, including *Aureobasidium* spp. and the related *Aspergillus* species for their functions and regulatory mechanisms. [9,10]. Previously published research studies have established that the types of TFs in *Aureobasidium* spp. were much lower than those in *Aspergillus* spp., *Saccharomyces cerevisiae*, and other eukaryotes in the TRANSFAC (http://gene-regulation.com/) (accessed on 20 January 2019) and the YEASTRACT+ (http://www.yeastract.com) databases [10,11,12,13,14,15]. However, due to the high genomic diversity of *Aureobasidium* spp., representative data in current databases are underrepresented. Especially, many transcription factors lack the KEGG Orthologs (KO) (https://www.kegg.jp/kegg/ko.html, accessed on 14 April 2022). Meanwhile, no publications on the detailed classification, regulatory mechanism, or function of TFs in the *Aureobasidium* genus have been published, and no tools exist to effectively integrate current biological knowledge or transcription regulation of *Aureobasidium* spp. with the keywords of TF terms submitted by users. This hinders the study of the transcription regulation mechanisms of this genus.

This study manually selected and categorized the TF families and constructed a complete gene set—the “*Aureobasidium* Transcription Factor Database (ATFDB)” to retrieve TFs and download the Hidden Markov Models (HMMs) based on user-supplied search keywords. In order to widely search orthologs against this database, each type of the TFs is trained into the HMM-based profile. Users can quickly retrieve information by entering the names of different transcription factors or the types of TFs. For example, if you type Msn2, you obtain the relevant information about the global transcription factor Msn2 in *Aureobasidium* spp., and if you type C2H2, you obtain a TFs list of all these types in *Aureobasidium* spp. Not only does the ATFDB provide a minimum gene set for TFs in the *Aureobasidium* genus, but it also has thorough detailed information on TFs sharing among different strains of *Aureobasidium* spp. We have established an integrated TFs repertoire in the *Aureobasidium* spp., which promotes the discovery of transcription regulation in distinct *Aureobasidium* species.

## 2. Methods

### 2.1. Genome Collection and Data Analysis

A total of 1,578,889 protein sequences from 146 strains of *Aureobasidium* spp. are retrieved and manually downloaded from the GeneBank database (https://www.ncbi.nlm.nih.gov/assembly/) (accessed on 27 March 2022). Additionally, the representative genomes of the pullulan-producing strain *Aureobasidium melanogenum* P16, four reference strains of *Aureobasidium* spp. and fifty strains of *Aureobasidium pullulans* from various sources have been categorized for further investigation (Appendix A) [16,17].

### 2.2. Homologues TFs Searches and Training

The TF genes from the genomes of *S. cerevisiae* and *Aspergillus* spp. are utilized as a reference for searching the publicly available, assembled genomes of the *Aureobasidium* strains in the NCBI database (https://www.ncbi.nlm.nih.gov/). To identify the TFs, we used BLAST 2.5.0+ and BioEdit version 7.0.9.0 to search for homologous TFs in the genomes of *Aureobasidium* spp. The Hidden Markov Models (HMM) of various TFs are trained through the HMMER 3.3.2 software by using default parameters. The PFAM families and super-families of the DNA-binding domains are estimated and sorted through the Pfam (http://www.pfam.org/) and the SUPERFAMILY 2.0 web server (http://supfam.org).

### 2.3. Function Annotation and Domain Analysis

All the TFs are annotated in the GhostKOALA, the Cluster of Orthologous Groups of proteins (COG), Pfam, the Non-Redundant Protein Sequence Database (NR) and the Swiss-Prot database as previously described [18,19,20]. Each of them is summarized in reference to the YEASTRACT+ database and the published references for inferring the functions. The domains of the TFs are exhibited through the TBtools v1.098652 software.

### 2.4. Compilation of the TF Genes

All of the TFs’ information and sequences are publicly available on the ATFDB, which may be searched using keywords. The TF names are directly utilized as inputs (multiple inputs available) in the ATFDB, which can be used to search for their features and accession numbers. There are 45 different types of TFs that can be obtained, each with all of their feature annotations applied manually (Appendix A). Furthermore, we make every attempt to ensure that gene sequences for each kind of TF are obtained and downloaded for additional research.

### 2.5. TFs Classification between the NCBI and Transcription Factor Database

The TF classification assignments in the ATFDB and NCBI databases are compared throughout the genomes of *Aureobasidium* spp. in the GenBank database [21,22]. The classification of various transcription factors and the involved metabolic pathways is conducted based on the public articles. The sankey diagram is built in R 3.0.1+ by using the networkD3 package (https://www.r-graph-gallery.com/).

### 2.6. Database Construction

We aimed to construct ATFDB for comprehensive, accurate and rapid analysis of the transcriptional regulation mechanisms in the environmental adaptability of *Aureobasidium* spp. Thus, the constructed database should contain the most comprehensive and precise TFs gene families based on the current knowledge. In addition, TFs and their homologues from multiple orthology databases should be included to reduce false-positive assignments. Therefore, a pipeline was developed for ATFDB and included the following steps (Figure 1).

## 3. Results and Discussion

### 3.1. PFAM Families of DNA-Binding Domains in Aureobasidium spp.

The PFAM database currently has 670 entries for the term “DNA binding” (Interpro: 1053) of the PFAM families. After text-mining-based filtering of the general TFs, the putative and basal metabolic proteins, 45 common PFAM families were finally identified and selected for analysis based on these family entries (Table 1). Additionally, 16 superfamilies were predicted among the total number of superfamilies of DNA-binding domains in reference to the type strain *Aureobasidium melanogenum* CBS110374. Among all these 16 superfamilies, 4 superfamilies were classified as specific transcription factors: the Zn2/Cys6 (Zn cluster), the DNA-binding domain of Mlu1-box-binding protein Mbp1, the basic-leucine zipper (bZIP) and the Zinc domain/Copper fist DNA binding domain (Table 1).

### 3.2. The Zn2Cys6 and Fungal-Specific TF Families

Four superfamilies and six PFAM families of the TFs were identified as the fungal-specific TFs in *Aureobasidium* spp. (Table 2). In fungi, these TFs are genus-specific and quite function-specific, regulating the fungal-specific morphogenetic processes that remain ubiquitous and conservative functions in *Aureobasidium* spp. The analysis results of the 45 fungal transcription regulator-related PFAM domains suggest that there are significant differences in the number of transcription regulators across different fungal classes (Figure 2 and Appendix A). The fungal specific transcription factor domain (PF04082) and the fungal Zn(2)-Cys(6) binuclear cluster domain (PF00172) comprise a large proportion in different fungi (Figure 2). The numbers of the two Pfam families were found to be larger in *Aspergillus* spp. and *Aureobasidium* spp. than that in *S. cerevisiae* (Appendix A). These findings indicate that since the *Aureobasidium* lineage separated, different classes of fungi have evolved different regulatory mechanisms.

In *S. cerevisiae*, the total number of the regulators is very low, which is consistent with the general PFAM distribution. Generally, the Zn2Cys6 and fungal-specific domain regulators have overwhelmingly predominated in *Aureobasidium* spp., whereas the C2H2 zinc finger class is a much lesser regulator class. However, compared with that in *S. cerevisiae*, the proportion of the C2H2 types decreased significantly in *Aureobasidium* spp. and *Aspergillus* spp. (Figure 2). Furthermore, the classification, function, and regulation of these fungal-specific TF families have been further elucidated as described below, including zinc cluster TF families, fungal-specific domain TF families, KilA-N domain TF families, HMG-box domain TF families, basic Leu zipper (bZIP) families and forkhead TF families.

#### 3.2.1. The Zinc Cluster TF Families

The Zn2/Cys6 (Zn cluster) superfamily (PF00172) is a vast group of the fungal-specific TFs that regulate many critical metabolic processes involved in cell growth and development [15]. Likewise, the Zn(II)_2_Cys_6_ domain family proteins in *Aureobasidium* spp. (numbers: 141) and *Aspergillus* (numbers: 222) are greater than that in *S. cerevisiae* (numbers: 48) (Figure 2), suggesting that the major expansion of these regulator classes may have occurred following the divergence of *Aureobasidium* spp. and *Aspergillus* spp. at ancient times [18]. For tuning iron uptake and storage of *Aureobasidium melanogenum* HN6.2, the transcriptome analysis demonstrates that the zinc cluster transcription factor SreA regulates and maintains iron homeostasis (Appendix A) [23].

#### 3.2.2. The “Fungal-Specific Transcription Factor Domain” TF Families

The second large class of TFs is the “fungal-specific transcription factor domain” (PF04082, IPR007219), which has expanded more rapidly in the Ascomycota than in the Basidiomycota [24]. In *Aureobasidium* spp., it consists of three members: the Fungal trans (PF04082), the Cep3 (PF16846), and the Fungal trans 2 (PF04082 or PF11951). Transcription factors with this domain are involved in the metabolic flow of carbon pool and fatty acids (Table 2).

#### 3.2.3. The KilA-N Domain TF Families

The KilA-N domain (PF04383) has been verified to regulate cellular differentiation, melanin synthesis, cell growth, and development in *Aureobasidium* spp. Meanwhile, ankyrins are multifunctional adaptors that link specific proteins to the membrane-associated, spectrin-actin cytoskeleton. The results show that although there are no differences in the KilA-N domain of Swi4/6 among different fungi, in *Aureobasidium* spp., transcription factor Swi4/6 has a long sequence coverage of the ankyrin repeats (ANK) with at least three consecutive copies, which is obviously different from those in *A. fumigatus* and *S. cerevisiae* (Figure 3) [25].

#### 3.2.4. The HMG-Box Domain TF Families

The HMG-box domain (PF09011), the homologues of MATalpha_HMGbox (PF04769, IPR006856) and the MAT_HMGbox (cd01389) can be clearly identified in each of the investigated *Aureobasidium* strains, such as the *MAT1-1-1* gene (QBQ58041.3) and the *MAT1-2-1* gene (QBQ58040.3) (Table 2) (Appendix A) [16].

#### 3.2.5. The HTH, Basic Leu Zipper (bZip) and Forkhead (FH) TF Families

The HTH, the GATA zinc finger, and the basic Leu zipper motif (bZIP) are also classified as yeast-specific TFs [26]. The βHLH regulator involves the homologue of Ino4 (KEQ67287.1) in *Aureobasidium* spp., which is equivalent to the regulator of phospholipid synthesis in *S. cerevisiae*. Sre1 (KEQ62711.1) and SreA (KEQ65560.1) are identified in the genome of *Aureobasidium* spp., of which they are transcription repressors that inhibit the biosynthesis of iron carriers in *Aureobasidium* spp. The bZIP proteins mainly are primarily involved with the iron acquisition regulator HapX in the synthesis of siderophore (Appendix A).

### 3.3. Identification of Transcription Factors in Aureobasidium spp.

The transcription factors of *Aureobasidium* spp. are identified by searching annotated genes and comprehensive results of biochemical and genetic analysis of *S. cerevisiae* and *Aspergillus fumigatus*. Generally, the homologous TFs found in *Aureobasidium* spp. are manually curated according to the literature of previous research (Appendix A). Forty-five transcription factors characterize and represent a variety of distinct functions in *Aureobasidium* spp., including six involved in the nitrogen metabolism (Seb1/Msn2, AreA, AreB, MeaB, TamA and Put3), five involved in synthesis of siderophores (AcuM, Sre1, SreA, HapX and Hap2), four involved in synthesis of polysaccharides (XlnR/GliZ, Gal4, AmyR and Ftr1), two involved in the glycolysis (Gcr1), and gluconeogenesis regulation (Cat8), while six general TFs are associated with RNA polymerase II transcription (TFⅡA, TFⅡB, TFⅡD, TFⅡE, TFⅡF and TFⅡH), and another six are involved in the signal pathway regulation, such as the HOG, cAMP-PKA, MAPK, and CWI pathways, as well as stress response pathway (Appendix A). Approximately three metabolic pathway-specific transcription factors are found to be associated with the regulation of secondary metabolites, such as the homologues transcriptional activator AflR, Cmr1/Pig1 and the GATA-type sexual development transcription factor NsdD (Appendix A).

Additionally, the other regulators mainly participate in the biochemical processes of phospholipid and fatty acid synthesis, chromatin remodeling, DNA repair, oxidative stress response, heat shock response, and protein folding (Appendix A). The transcriptional activator Hsf1 (KEQ66690.1), for example, can specifically recognize the nGAAn repeat units in the promoter region of heat shock elements (HSEs), therefore regulating heat shock proteins for thermotolerance [27]. It is widely distributed in the genomes of *Aureobasidium* spp. (Appendix A). However, most of the homologues involved in the carbon metabolism (GalX, GalR, ScfA, AlcR, AceII) are found to be lost in *Aureobasidium* spp., indicating a genus specificity in *Aspergillus* spp.

#### 3.3.1. Taxonomy and Functions of the Zinc Finger TFs in *Aureobasidium* spp.

The Zinc finger (PFAM entries: 257) exists in various proteins and has a wide range of functions in various cellular processes [28]. There are 215 genes encoding zinc finger proteins in the genomes of *Aureobasidium* spp., the number of which is comparable to that of *S. cerevisiae* (217) and *Aspergillus niger* (2951). Initially, zinc fingers were categorized according to the different numbers and orders of the residues (Cys2His2, Cys4, and Cys6). Krishna et al. [28] previously described a more systematic approach for classifying zinc finger proteins into “fold groups” based on the overall structure of the protein backbone. The PFAM database presently contains 257 zinc finger families; thus, after manually filtering the unrelated and unknown functional proteins, we finally summarized the identified zinc finger families into seven “fold groups” in *Aureobasidium* spp. as follows (Appendix A). Seven types of zinc finger transcription factors (class Ⅰ-Ⅶ) have been found in the genomes of *Aureobasidium* spp. (Appendix A), which are widely involved in the regulation of osmotic pressure regulation, secondary metabolites, the MAPK pathway, the calcium signaling pathway and glucose suppression (Appendix A).

#### 3.3.2. Classification and Functions of the C2H2 TFs in *Aureobasidium* spp.

The C2H2 and C2HC motifs are highly conserved in the majority of fungi, and they are involved in transcription regulation [9]. We searched for the C2H2 TFs in the genomes of *Aspergillus aculeatus* ATCC16872, *A. melanogenum* P16, *A. melanogenum* CBS110374, *Aureobasidium subglaciale* EXF-2481, *Aureobasidium pullulans* EXF-150 and *Aureobasidium namibiae* CBS147.97, and the neighbor-joining (NJ) phylogenies for the datasets of sequences were then inferred by MEGA v7.0.26. According to the different domains and functions of the C2H2 TFs, the homologues can be divided into eight categories in different strains of *Aureobasidium* spp. (Figure 4A), which mainly contain the global transcription factors, specifically TFs, and metabolic pathways of specific transcription factors (Appendix A).

Global transcription factors are classified into four groups (groups 1, 4, 5 and 6), such as the transcription repressor CreA/Mig1 (KEQ62800.1), the transcriptional activator Seb1/Msn2 (KEQ58292.1), Crz1 (KEQ62890.1) and Ste12/BrlA (KEQ57897.1) (Figure 4A). In *Aureobasidium* spp., CreA is known to be important in the regulation of pullulan biosynthesis, and deletion of the *CREA*/*MIG1* gene causes de-repression of the expression of several genes in *A. melanognum* P16 under high-concentration glucose medium [29]. Seb1/Msn2, a global transcription factor, acts as a regulator in the osmotic pressure response pathway to glycogen accumulation. The cellular localization of Msn2 is determined by phosphorylation, and the dephosphorylated Msn2 is clustered in the cell nuclei to activate the expression of the *UGP1* gene and regulates pullulan production in *A. melanogenum* P16 [7,30]. Crz1 commonly regulates gene transcription in response to environmental changes through dephosphorylation by a Ca^2+^/calmodulin-dependent phosphatase [31]. PMA biosynthesis is regulated by the transcriptional activator Crz1 from the Ca^2+^ signaling pathway [32,33], and this transcriptional activator can be found in most *Aureobasidium* genomes (Appendix A). The global transcription factor Ste12 and Ste12-like proteins (KEQ57897.1) are found exclusively in fungi that are activated by a cascade of mitogen-activated protein kinase (MAPK) signals. The C-terminal C2H2-Zn^2+^ finger domains are found in *Aureobasidium* spp. and *Aspergillus* spp. but not in *S. cerevisiae*, *Yarrowia lipolytica*, *Kluyveromyces marxianus* and *Candida albicans* [32,33]. Fungal-specific transcription factors contain two groups (groups 2 and 3), such as the PacC (KEQ63386.1) and Mac1 (KEQ60235.1) (Figure 4A). PacC is activated by a signal transduction pathway triggered by neutral or alkaline pH [34]. Another one, the fungal-specific transcription factor Mac1, is indispensable to switches of gene expression required for high affinity copper ion transport [35,36]. Metabolic pathway-specific transcription factors are also classified into two groups (groups 7 and 8), including ScpR/AfoA (KEQ63975.1) and Cmr1/Pig1 (KEQ67497.1) (Figure 4A) [25,37]. There are special amino acid residues of DNA interaction—R, H, R and R, E, R in the C2H2 Zn finger of *Aureobasidium* spp. (Figure 4B). ScpR/AfoA can regulate the aspartate gene cluster, *NRPS* gene (*inpA* and *inpB*) and biosynthesis of asper furanone expression [35]. In fungi, the *PKS**1* gene and *C**MR1* gene links together and constitutes a gene cluster to catalyze melanin synthesis. The metabolic pathway-specific transcription factor Cmr1 can regulate the expression of the *PKS* gene in *Aureobasidium* spp. [25]. Cmr1 has three types of functional motifs as mentioned above (Figure 5), indicating that it is a unique metabolic pathway specific transcription factor. It has similar domains in the proximal species and may only exist in most asexual lineages of Dothideomycetes, such as *Aureobasidium* spp. and other melanin-producing fungi (Figure 5). In contrast, it was found to be lost in the *Aspergillus* of Ascomycete and is only confined to part of the melanin-producing fungi, indicating that Cmr1 is also a species-specific transcription factor [25] (Figure 5).

### 3.4. Transcription Factor Database for Aureobasidium spp.

To establish a complete gene set of TFs in the *Aureobasidium* genus, we investigated the reported transcription factors in the published genomes of *Aureobasidium* spp. Finally, we identified a total of 5494 common homologous TFs in 1,578,889 proteins from 146 strains of *Aureobasidium* spp. using the reference strain *A. melanogenum* CBS110374. The identity thresholds and gene accessions can be checked in Appendix A. Further, the intact Hidden Markov Models (HMM) of different TFs have been perfectly trained and uploaded to the *Aureobasidium* Transcription Factor Database, which can be utilized to find more orthologous or paralogous TFs in this genus (see the HMMER 3.3.2 usage). This is more accurate than directly using the type strain for sequence alignment, because the genomes of type strains such as *A. melanogenum* CBS110374 or *A. pullulans* EXF-150 are the draft genomes with many genes to be incompletely assembled. In addition, the TFs in each of the fifty *A. pullulans* strains are chosen as the representative gene set for analysis since *A. pullulans* is the common species of *Aureobasidium* [16,17].

After submitting all the data, the *Aureobasidium* Transcription Factor Database has been established for TF retrieval. The database involves four sections and works on an intuitive approach: (i) users input the retrieval data; (ii) automatic word matching; (iii) query by exactly matching the database; (ⅳ) process and encapsulate the retrieved data format; (ⅴ) return the encapsulated data and render the returned data to show the results (Figure 6). All the data in the ATFDB is publicly accessible at https://huang.zgsj1.com/picture/literature/distweb/index.html (accessed on 15 April 2022), which also supports multiple inputs for users to retrieve. Meanwhile, all of the TFs can be clicked for information linking to external databases, and the KEGG entries for TFs can also be checked in the KEGG section. The user-friendly web server interface accepts user input and produces results in seconds.

We tested the ATFDB on a small dataset of the known C2H2 TFs. For example, when users search for the term “Crz1”, the detailed contents of the message are shown, including the transcription factors’ accession numbers and information relating to other databases. Furthermore, when entering the term “C2H2”, users can quickly retrieve the common transcription factors of this type, such as the transcription activator Cmr1 and the transcription repressor Mig1/CreA (Appendix A). This may also be used to search for various TFs in *Aureobasidium* spp., such as the Zn2Cys6, bZIP and GATA zinc fingers, demonstrating the database’s comprehensiveness and usefulness (Appendix A). Additionally, we provided the representative protein sequences of transcription factors from each of the fifty *A. pullulans* strains for users to download for further study. The ATFDB exhibits the known transcription regulation and gene associations of several TFs, highlighting its ability to contribute to the formulation of novel biological hypotheses. It is capable of identifying a significant number of candidate transcriptional regulators in *Aureobasidium* spp., many of which have been verified and studied in experiments [2,3,7,13,14,38,39,40,41]. Due to the increasing number of genome sequencing studies, no other approach, including YEASTRACT+, the NCBI, or the TRANSFAC database, could be used to obtain the species-specific TFs of *Aureobasidium* spp. The ATFDB can meet the needs of users by leveraging information about prior biological knowledge and phenotype to expedite and announce scientific discoveries.

### 3.5. Advantages of the ATFDB over Other Orthology Databases

In order to compare the differences between the NCBI and the ATFDB databases, we searched in NCBI through keywords according to different types of transcription factors as described above, such as “Msn2 *Aureobasidium*”. We then classified them according to the categories as stated above (including global, general and specific TFs) and counted the number of TFs contained within different categories. The results showed that in the type-level of TF classification, the NCBI (GenBank) database mostly contained general TFs (2511), but the ATFDB largely compensated for the NCBI (GenBank) database’s faults and weaknesses in the global TFs (1386) and particular TFs (697) (Figure 7A) (Appendix A). In particular, many TFs in the NCBI database are termed as hypothetical or putative proteins and are not further classified, which is inconvenient for the study of gene diversities. Apart from the fundamental metabolisms, the ATFDB outperforms the NCBI (GenBank) database in the signal pathways (426), secondary metabolisms (589), and certain other metabolic reaction processes (369) (Figure 7B) (Appendix A), suggesting that it complements and covers the majority of the common transcription factors. The classification of TFs and their participating metabolic pathways in the ATFDB are evaluated by comparison with that of the NCBI (GenBank) database.

## 4. Conclusions

The lineage of *Aureobasidium* spp. can synthesize a variety of natural compounds with high-economy value and potential industrial applications, the majority of which are associated with transcription factor metabolic regulatory networks. The built ATFDB can be applied for multiple purposes. Genes involved in each pathway are comprehensively and accurately presented in ATFDB. First, it holds the potential to be further used for genomic annotation. For example, if all transcriptional regulations involved in one pathway are annotated, we can consider the mechanisms of metabolism potentials of this pathway. In addition, ATFDB may be used for analyzing amplicon sequencing data of functional genes to separate homologous groups. In summary, this study presents a manually constructed database (ATFDB) that aims to profile transcription factor metabolic regulatory networks. It is publicly available without login requirements at https://huang.zgsj1.com/picture/literature/distweb/index.html (accessed on 14 April 2022). The ATFDB contains 16 superfamilies, 45 PFAM families of TFs with DNA-binding domains, 7 zinc finger families, and 8 categories of the C2H2-type TFs. The results demonstrate that ATFDB is a useful tool for studying the transcriptional regulations of *Aureobasidium* spp. in the environment, and it will be continuously updated.

## Figures and Tables

**Figure 1 jof-08-01096-f001:**
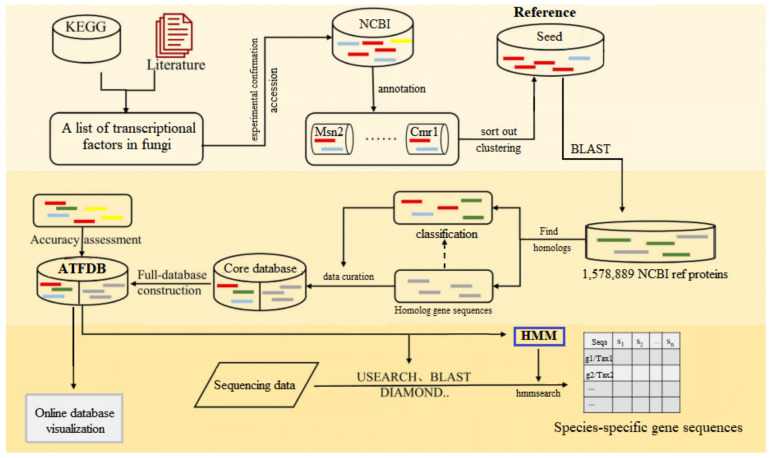
Flowchart of major steps of ATFDB construction. First, a seed database was constructed for selected genes by retrieving protein sequences from the NCBI database using the accession numbers. Second, target genes from the NCBI and their homologs were identified and integrated to construct the core database. At last, all data were uploaded to the online database for visualization, and the trained HMM models were developed to conduct species-specific TFs profiling of *Aureobasidium* spp. via the sequencing data.

**Figure 2 jof-08-01096-f002:**
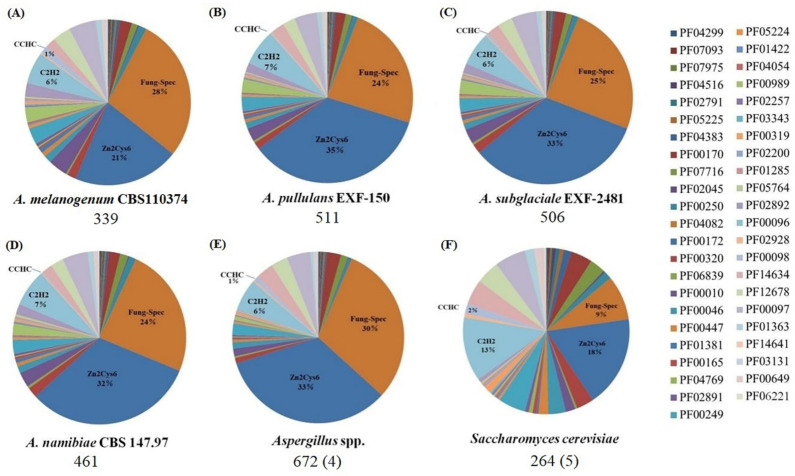
Distribution of PFAM families in different fungi. (**A**) *A. melanogenum* CBS110374; (**B**) *A. pullulans* EXF-150; (**C**) *A. subglaciale* EXF-2481; (**D**) *A. namibiae* CBS 147.97; (**E**) *Aspergillus* spp.: *A. aculeatus*, *A. niger*, *A. fumigatis* and *A. oryzae* RIB40; (**F**) *S. cerevisiae*: *S. cerevisiae* S288C, *S. cerevisiae* YJM244, *S. cerevisiae* YJM450, *S. cerevisiae* YJM993 and *S. cerevisiae* YJM1078. The average number of the regulators for each species is indicated underneath each pie chart. The number of genomes analyzed in all species is indicated in parentheses.

**Figure 3 jof-08-01096-f003:**
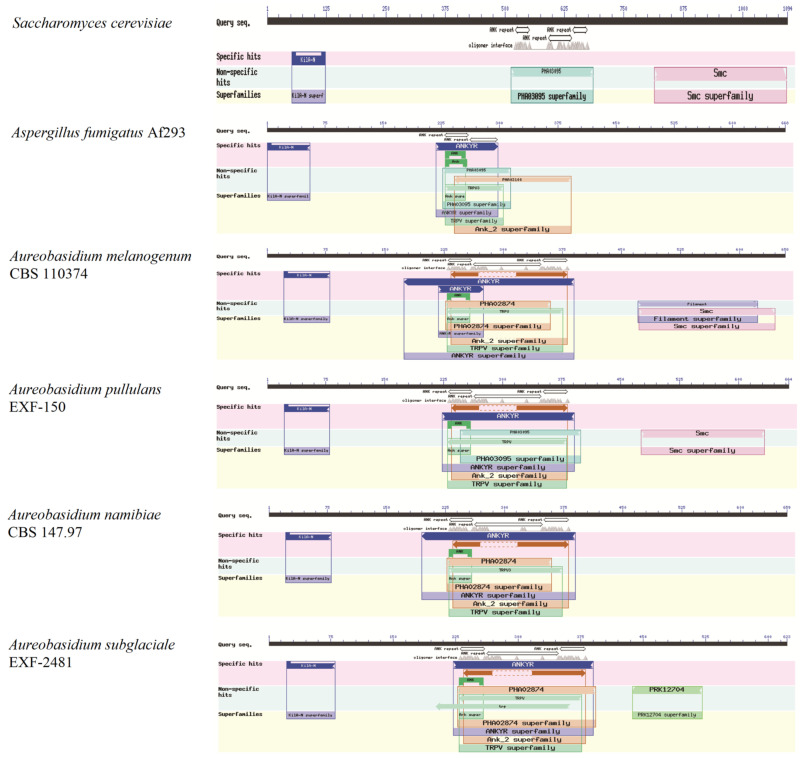
Conserved functional domains of the KilA-N type protein Swi4.

**Figure 4 jof-08-01096-f004:**
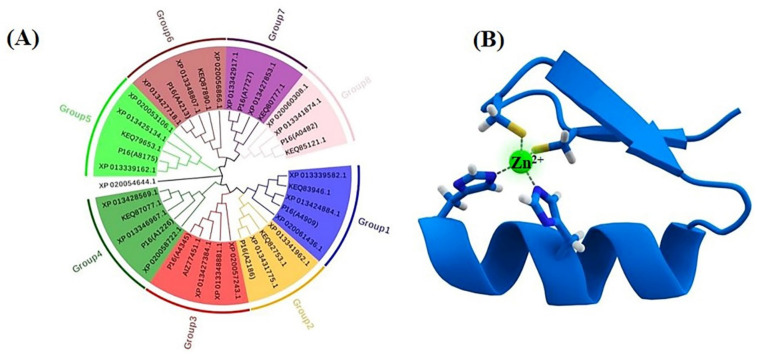
Phylogenetic tree of the C2H2 transcription factors (**A**) and the Cys2His2 zinc finger motif, consisting of α helix and an antiparallel β sheet (**B**). The zinc ion (green) is coordinated by two histidine residues and two cysteine residues.

**Figure 5 jof-08-01096-f005:**
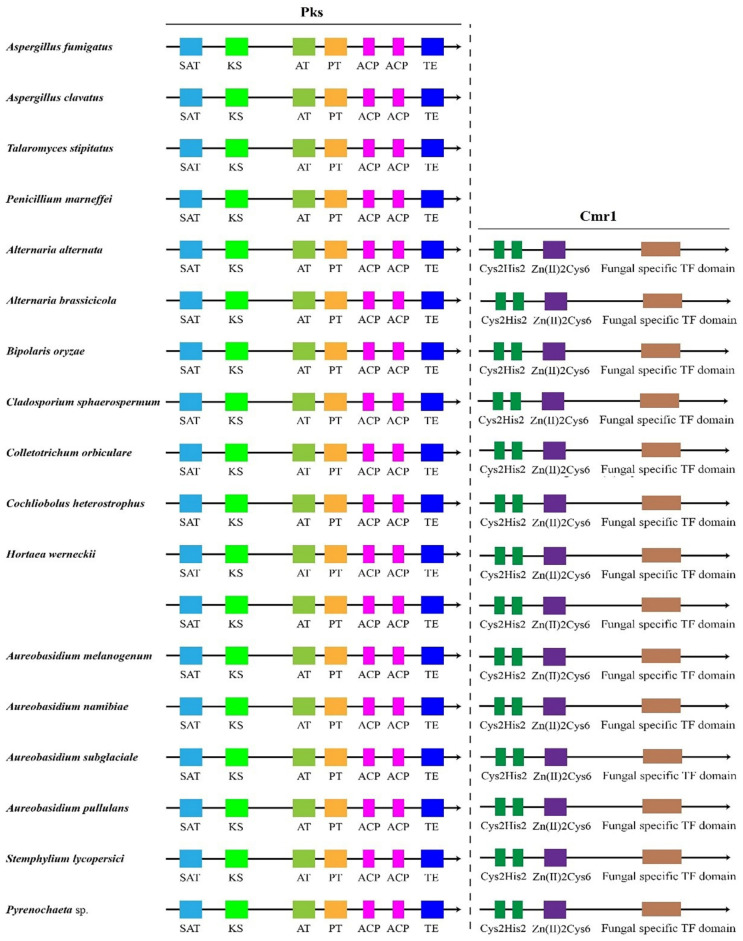
Transcription factor Cmr1 and the target protein Pks1 in different fungi.

**Figure 6 jof-08-01096-f006:**
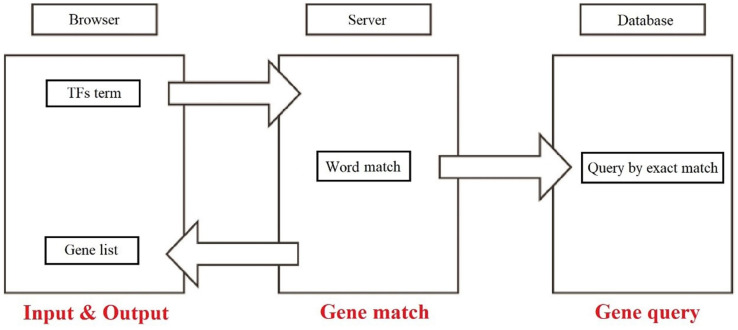
Workflow of the *Aureobasidium* Transcription Factor Database.

**Figure 7 jof-08-01096-f007:**
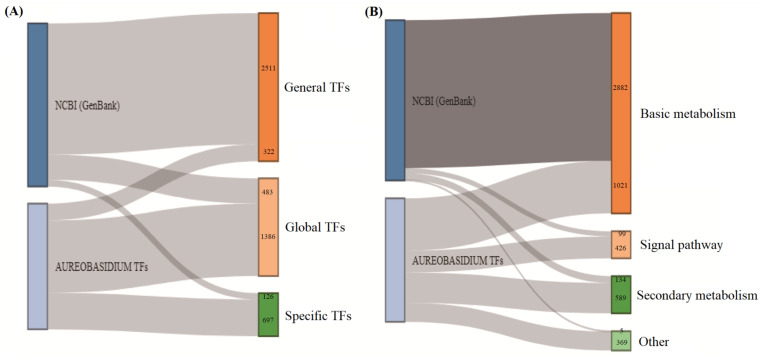
Comparison of the TFs numbers of *Aureobasidium* spp. in the NCBI (GenBank) and the ATFDB databases. (**A**) Comparison of the NCBI (GenBank) and the ATFDB databases at type-level of TFs classifications. (**B**) Comparison of the NCBI (GenBank) and the ATFDB databases at the metabolism-level of TFs classifications.

**Table 1 jof-08-01096-t001:** Sixteen superfamilies and forty-five PFAM families of TF DNA binding domains predicted to occur in *Aureobasidium* spp. (except for general transcription factors, DNA repair/extend, nuclear receptor and chromatin remodeling).

PFAM Family	PFAM ID
KilA-N domain	PF04383
bZIP transcription factor (bZIP_1)	PF00170
Putative FMN-binding domain	PF04299
SGT1 protein	PF07093
TFIIH C1-like domain	PF07975
CP2 transcription factor	PF04516
DDT domain	PF02791
Helix–turn–helix, Psq domain	PF05225
Basic region leucine zipper 2	PF07716
CCAAT-binding TF (CBF-B/NF-YA) subunit B	PF02045
Forkhead domain	PF00250
Fungal specific transcription factor domain	PF04082
Fungal Zn(2)-Cys(6) binuclear cluster domain	PF00172
GATA zinc finger	PF00320
GRF zinc finger	PF06839
Helix-loop-helix DNA-binding domain	PF00010
Homeodomain (Homeobox)	PF00046
HSF-type DNA-binding	PF00447
Helix-turn-helix (HTH_3)	PF01381
Bacterial regulatory helix-turn-helix proteins, AraC family	PF00165
Mating-type protein MAT alpha 1 HMG-box	PF04769
MIZ/SP-RING zinc finger	PF02891
Myb-like DNA-binding domain	PF00249
NDT80/PhoG-like DNA-binding family	PF05224
NF-X1 type zinc finger	PF01422
CCR4-Not complex component, Not1	PF04054
PAS fold	PF00989
RFX DNA-binding domain	PF02257
SART-1 family	PF03343
SRF-type TF (DNA-binding and dimerization domain)	PF00319
STE-like transcription factor	PF02200
TEA/ATTS domain	PF01285
YL1 nuclear protein	PF05764
BED zinc finger	PF02892
Zinc finger, C2H2 type	PF00096
Zinc finger, C5HC2 type	PF02928
Zinc knuckle(zf-CCHC)	PF00098
Zinc-Ring finger domain	PF14634
Ring-H2 zinc finger domain	PF12678
Zinc finger, C3HC4 type (Ring finger)	PF00097
FYVE zinc finger	PF01363
Helix-turn-helix DNA-binding domain of SPT6	PF14641
bZIP Maf transcription factor	PF03131
Copper fist DNA binding domain	PF00649
Putative zinc finger motif, C2HC5-type	PF06221
**Superfamily**	**Superfamily ID**
beta-beta-alpha zinc fingers (C2H2 and C2HC zinc fingers)	57667
DNA-binding domain of Mlu1-box-binding protein Mbp1	54616
Glucocorticoid receptor-like (DNA-binding domain)	57716
Helix-loop-helix DNA-binding domain	47459
Homeodomain-like	46689
Lambda repressor-like DNA-binding domains	47413
Nucleic acid-binding proteins	50249
p53-like transcription factors	49417
SRF-like	55455
Winged helix DNA-binding domain	46785
Zinc domain conserved in yeast copper-regulated TFs	57879
Zn2/Cys6 DNA-binding domain	57701
HMG-box	47095
Putative DNA-binding domain	46955
basic-leucine zipper (bZIP)	57959
WD40 repeat-like	50978

**Table 2 jof-08-01096-t002:** The Zn2Cys6 and fungal-specific TFs in *Aureobasidium* spp.

Types	Family	Function
Zn2/Cys6 (Zn cluster)	Superfamily: 57701	Galactose metabolism, proline utilization, nitrogen metabolism, sugar and amino acid metabolism, polysaccharide metabolism, ironophore biosynthesis, xylan/cellulose degradation, amylohydrolysis, regulation of secondary metabolite clusters.
DNA-binding domain of Mlu1-box-binding protein Mbp1	Superfamily: 54616	Cell cycle regulation
Basic-leucine zipper (bZIP)	Superfamily: 57959	Metabolism of iron element, osmotic and oxidative stress reactions, regulation of HOG pathway, chromatin remodeling and nitrogen metabolism.
Zinc domain conserved in yeast copper-regulated TFs/Copper fist DNA binding domain	Superfamily: 57879 PFAM: PF00649	Activation of the transcription of the metallothionein gene in response to copper
Fungal-specific transcription factor domain	PFAM: PF04082	Sugar and amino acid metabolism, fatty acid catabolism and other cellular and metabolic processes
KilA-N domain	PFAM: PF04383	Cell cycle, pseudohyphal differentiation, morphogenesis and metabolism.
Mating-type protein MAT alpha 1 HMG-box	PFAM: PF04769	Activation of mating-type α-specific genes, bind cooperatively with Mcm1 to PQ elements upstream of α-specific genes
Helix-turn-helix (HTH_3)	PFAM: PF01381	Regulation of RNA transcription
Helix-loop-helix DNA-binding domain	PFAM: PF00010	Conidiospore regulation and pseudohyphal differentiation, phospholipid and fatty acid synthesis, chromatin remodeling.
Forkhead domain	PFAM: PF00250	Chromatin silencing at HML and HMR, regulation of the G2/M phase gene expression

## Data Availability

All the data are incorporated into the article and the *Aureobasidium* Transcription Factor Database (https://huang.zgsj1.com/picture/literature/distweb/index.html, accessed on 15 April 2022). The data underlying this article are available in the article and in its online supplementary material.

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
