# Peer review of "Transcription Factors in Aureobasidium spp.: Classification, Regulation and a Newly Built Database"

_jof, 2022, doi:10.3390/jof8101096_

Round 1
Reviewer 1 Report
In the manuscript the authors made a database for the transcription factors in Aureobacidium spp. It was valuable for the study of metabolic regulation of this fungus in near future. Although the most of this study was carried out by the "dry" work, the manuscript contains many useful information. Therefore I recommend the manuscript for publication in JoF.
Author Response
We are deeply grateful for your recognition of this research, which gives us more confidence to continue this study. Thanks for taking your time to review this manuscript again.
Reviewer 2 Report
Dear Authors,
I found your article as interesting, there is always a need to develop new pipelines which help other authors organize their data.
Author Response

(The authors gave the same response as above.)

Reviewer 3 Report
The manuscript "Transcription factors in Aureobasidium spp.: classification, regulation and a newly built database" introduces a manually curated resource - a database of transcription factors in Aureobasidium spp. This database is working useful resource for practitioners. The pipeline of the database creation is also explained in the manuscript and maybe useful methodological resource to guide a creation of similar databases for other less characterized species.
Minor comments are provided below.
L 50 - language and expression consistency - TFs were much lower- what do you mean - a number of TFs?
L 63 - please explain what HMM models are available in your database and what the user-supplied keywords designate. Provide an example of such keyword and describe the area that these keywords cover.
Figure 1 - comparison between Aspergillus and Aureobasidiom and S.cerevisiae - please add different figure of either bar chart or dot plot to represent exact numbers of the PFAM families that you wish to distinguish. The pie chart representation obscure the comparison.
Figure 2 - Please provide and explain the data source used to represent the distribution of domains in different species. What is ANK and Blast:ANK ? Please explain. You are saying that there is one more ANK domain in Aureobasidium and refer to the Figure 2, but in each organism there are 2 ANK domains ?.. Or you mean Blast:ANK? Please explain what is exactly ANK and Blast:ANK im your figure.
L 155 - please correct language " Identification " or "Identified transcription factors"
Figure 4 - Please explain the Pks and Cmrl in more detail. What Figure 4 is supposed to communicate?
Line 291, Paragraph 2.5 - it is not clear how exactly the comparisons between the NCBI GeneBank and ATFDB were performed. Please provide more detailed description of how the comparisons were carried out and what exactly the ribbons in the Figure 6 represent. It would help to see the exact numbers in the figure that the ribbons represent.
I would also recommend that native English speaker would proofread the manuscript, because there are some poor English expressions. Correcting them would improve this manuscript substantially.
Author Response
We are very grateful for your helpful comments on our manuscript. We revised the manuscript following your advice. Modified portions are marked in red on the paper. Here below is a one-by-one response to your comments.
